# Which MOoD Methods work?
# A Benchmark of Medical Out of Distribution Detection

**Author(s) names withheld**                                             EMAIL(S) WITHHELD

## Abstract

There is a rise in the use of deep learning for automated medical diagnosis, most notably in medical imaging. Such an automated system uses a set of images from a patient to diagnose whether they have a disease. However, systems trained for one particular domain of images cannot be expected to perform accurately on images of a different domain. These images should be filtered out by an Out-of-Distribution Detection (OoDD) method prior to diagnosis. This paper benchmarks popular OoDD methods in three domains of medical imaging: chest x-rays, fundus images, and histology slides. Our experiments show that despite methods yielding good results on some types of out-of-distribution samples, they fail to recognize images close to the training distribution.

**Keywords:** Medical imaging, out-of-distribution detection, chest X-ray, fundus, histology.

## 1. Introduction

A safe system for medical diagnosis should withhold diagnosis on cases outside its validated expertise. For machine learning (ML) systems, the expertise is defined by the validation score on the distribution of data used during training, as the performance of the system can be validated on samples drawn from the same distribution (as per PAC learning (Valiant, 1984)). This restriction can be translated into the task of *Out-of-Distribution Detection* (OoDD), the goal of which is to distinguish between samples in and out of a desired distribution (abbreviated to *In* and *Out* data). In this case, *In* data is the training distribution of the diagnosis system.

Manual design of an OoDD system is difficult because it requires modeling the desired and undesired distributions of data. The later is widely varying and often unseen until deployment. For example, an OoDD system for frontal chest X-ray images needs to reject lateral view images, non-chest X-ray images, and images that had an error during acquisition. The variability of data outside the desired distribution makes hand-engineering OoDD systems with hand-designed features extremely difficult and motivates the use of machine learning based OoDD systems.

In contrast to natural image analysis, medical image analysis must often deal with orientation invariance (e.g. in cell images), high variance in feature scale (in Xray images), and locale specific features (e.g. CT) (Razzak et al., 2017). A systematic evaluation of OoDD methods for applications specific to medical image domains remains absent, leaving practitioners blind as to which OoDD methods perform well and under which circumstances. This paper fills this gap by benchmarking many current notable OoDD methods in three medical

image tasks as *In* data distributions: chest X-ray, fundus, and histology imaging. In our study, we evaluate each OoDD method under three *Out* data distributions (called usecases) which are different types of deviations from the *In* data. Our procedure contributes to the understanding of challenges faced in OoDD under various situations faced by systems built around medical images.

OoDD and its relationship to generalization is a challenging problem that can be thought about from multiple perspectives (Nalisnick et al., 2019; Ahmed & Courville, 2019; Metzen et al., 2017; Lee et al., 2018; Chalapathy & Chawla, 2019). Many approaches validate their theory on common ML datasets which can leave us blind to failure modes that exist in real life settings. Our empirical studies show that current OoDD methods perform poorly when detecting correctly acquired images that are not represented in the training data (and therefore would yield inexpected results). We also find that in other usecases we have a different conclusion than the prior work of Shafaei et al. (2018), which benchmark the OoDD methods on a suite of natural image datasets.

## 2. Task Formulation

We identify three distinct out-of-distribution categories, justified below:

- **usecase 1** Reject inputs that are unrelated to the evaluation. This includes obviously-wrong inputs from a different domain (e.g. fMRI image in X-ray, cartoon in natural image etc) and less obviously-wrong inputs (e.g. wrist X-ray in chest X-ray).

- **usecase 2** Reject inputs which are incorrectly prepared (e.g. blurry image of chest X-ray, poor contrast, Lateral vs Dorsal position).

- **usecase 3** Reject inputs that are unseen due to a selection bias in the training distribution (e.g. image with an unseen disease).

We justify these usecases by enumerating different types of mistakes or biases that can occur at different stages of the data acquisition. This is visually represented in Figure 1. As each usecase could be individually relevant to a specific application, we will evaluate OoDD methods for satisfying each requirement individually.

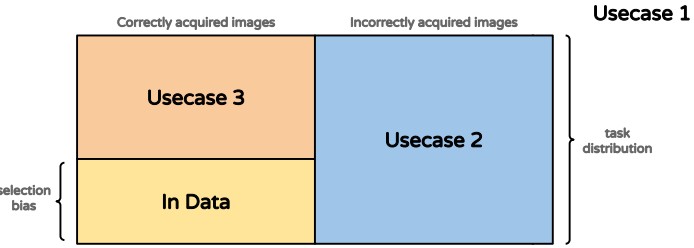

Figure 1: The three usecases shown in relation to each other. The training data is sampled iid from the *In* data distribution.

**OD-test Evaluation Framework** We follow the "OD-test" framework proposed in Shafaei et al. (2018), except that we use multiple datasets for calibration (explained below) to have a better estimate for *Out* data.

First, a model of the *In* data distribution is learned on a split of *In* data which we denote as $D_{tr}$. Secondly, a OoDD classifier is calibrated using a set denoted as $D_{val}$ which requires both *In* and *Out* data. Finally, the performance of OoDD is evaluated using a third split $D_{test}$ which again requires both *In* and *Out* data. A random three way split for *In* data and a random two way split for *Out* data are used.

While splitting the *In* data is generally simple for most datasets in practice, splitting the *Out* data requires more consideration to avoid using the same set of data appearing in the $D_{val}$ split for the $D_{test}$ split, which tends to result in an overestimation of performance. To this end, Shafaei et al. (2018) proposes a more realistic evaluation where disjoint *Out* datasets are used in $D_{val}$ and $D_{test}$.

## 3. Methods of OOD Detection

We consider three classes of OoDD methods. Data-only methods do not rely on any pre-trained models and are learned directly on $D_{val}$. Classifier-only methods assume access to a downstream classifier trained for classification on *In* data ($D_{tr}$). Methods with auxiliary models requires pre-training of a neural network that is trained on *In* data through other tasks such as image reconstruction or generative modeling. The threshold values of the methods below are all calibrated on a subset of $D_{val}$.

**Data-only methods**  The most simple and easy to implement baseline is k-Nearest-Neighbors (KNN) which only needs to observe the training data. This is performed on images as a baseline for our evaluations. For speed only 1000 samples are used from $D_{tr}$ to calculate neighbor distance. A threshold is determined using samples from $D_{val}$.

**Classifier-only methods**  Classifier-only methods make use of the downstream classifier for performing OoDD. Compared to data-only methods they require less storage, however their applicability is constrained to cases with classification as downstream tasks. *Probability Threshold* (Hendrycks & Gimpel, 2017) uses a threshold on the prediction confidence of the classifier to perform OoDD. *Score SVM* trains an SVM on the logits of the classifier as features, generalizing probability threshold. *Binary Classifier* trains on the features of the penultimate layer of the classifier. *Feature KNN* uses the same features as the binary classifier, but constructs a KNN classifier in place of logistic regression. *ODin* (Liang et al., 2017) is a probability threshold method that preprocesses the input by taking a gradient step of the input image to increase the difference between the *In* and *Out* data. *Mahalanobis* (Lee et al., 2018) models the features of a classifier of *In* data as a mixture of Gaussians, preprocesses the data as ODin, and thresholds the likelihood of the feature.

**Methods with Auxiliary Models**  OoDD methods in this section require an auxiliary model trained on *In* data. This results in extra setup time and resources when the downstream classifier is readily available. However, this could also be advantageous when the downstream task is not classification (such as regression) where methods may be difficult to adapt. *Autoencoder Reconstruction* thresholds the reconstruction loss of the autoencoder to achieve OOD detection. Intuitively, the autoencoder is only optimized for reconstructing *In* data, and hence reconstruction quality of *Out* data is expected to be poor due to the bottleneck in the autoencoder. In this work we consider three variants of autoencoders: standard autoencoder (AE) trained with reconstruction loss only, variational autoencoder trained

| Domain | Eval | *In* data | Usecase 1 *Out* data | Usecase 2 *Out* data | Usecase 3 *Out* data |
|---|---|---|---|---|---|
| Chest X-ray | 1 | NIH (*In* split) | UC-1 Common, MURA | PC-Lateral, PC-AP, PC-PED, PC-AP-Horizontal | NIH-Cardiomegaly, NIH-Nodule, NIH-Mass, NIH-Pneumothorax |
| | 2 | PC-Lateral (*In* split) | UC-1 Common, MURA | PC-AP, PC-PED, PC-AP-Horizontal, PC-PA | PC-Cardiomegaly, PC-Nodule, PC-Mass, PC-Pneumothorax |
| Fundus Imaging | 3 | DRD | UC-1 Common | DRIMDB | RIGA |
| Histology | 4 | PCAM | UC-1 Common, Malaria | ANHIR, IDC | None |

Table 1: Datasets used in evaluations. See Appendix B for more details.

with a variational lower bound (VAE) (Kingma & Welling, 2014), and decoder+encoder trained with an adversarial loss (ALI (Dumoulin et al., 2016), BiGAN (Donahue et al., 2017)). Furthermore, we include two different reconstruction loss functions in the benchmark: mean-squared error (MSE) and binary cross entropy (BCE). Finally, *Autoencoder KNN* constructs an KNN classifier on the features output by the encoder.

## 4. Experiment Setup

Here the four evaluations are explained. See Table 1 for a summary and Appendix A for more detailed description of the datasets.

In *evaluation 1*, *In* data is frontal chest X-ray images. The task is to predict 10 of the 14 radiologcal findings defined by the **NIH** Chest-X-Ray14 dataset (Wang et al., 2017). The remaining labels become usecase 3.

In *evaluation 2*, *In* data is lateral chest X-ray images. The task is the same as evaluation 1, but the *In* data is from lateral view images in the PADChest dataset (Bustos et al., 2019).

In *evaluation 3*, *In* data is fundus/retinal (back of the eye) images. The task is to predict if diabetic retinopathy is present in the image defined by the **DRD** (Diabetic Retinopathy Detection) [1] dataset. Here usecase 3 is images with glaucoma and not diabetic retinopathy (Almazroa et al., 2018).

In *evaluation 4*, *In* data is H&E stained histology slides of lymph nodes. The task is to predict if image patches contain metastatic (cancerous) tissue defined by the **PCAM** dataset (Veeling et al., 2018). Here usecase 3 is not included.

All prerequisite networks for each OoDD method are trained to convergence on $D_{tr}$ of each evaluation with a held-out split for validation. When training for usecase 1, three *Out* datasets are randomly selected for $D_{val}$ while the rest is used for $D_{test}$. For usecases 2 and 3, we enumerate over configurations where each *Out* dataset is used as $D_{val}$ with the rest as $D_{test}$. $D_{val}$ and $D_{test}$ are class-balanced by subsampling equal numbers of *In* and *Out* samples. Hyperparameter sweep is carried out where needed. 10 repeated trials, with re-sampled $D_{val}$ and $D_{test}$, are performed for each evaluation.

We measure the accuracy and Area Under Precision-Recall Curve (AUPRC) of all methods on each $D_{test}$. Since $D_{test}$ is class-balanced, accuracy provides an unbiased representation of type I and type II errors. AUPRC characterizes the separability of *In* and *Out* samples in predicted value (the value that we threshold to obtain classification).

---

1. https://www.kaggle.com/c/diabetic-retinopathy-detection

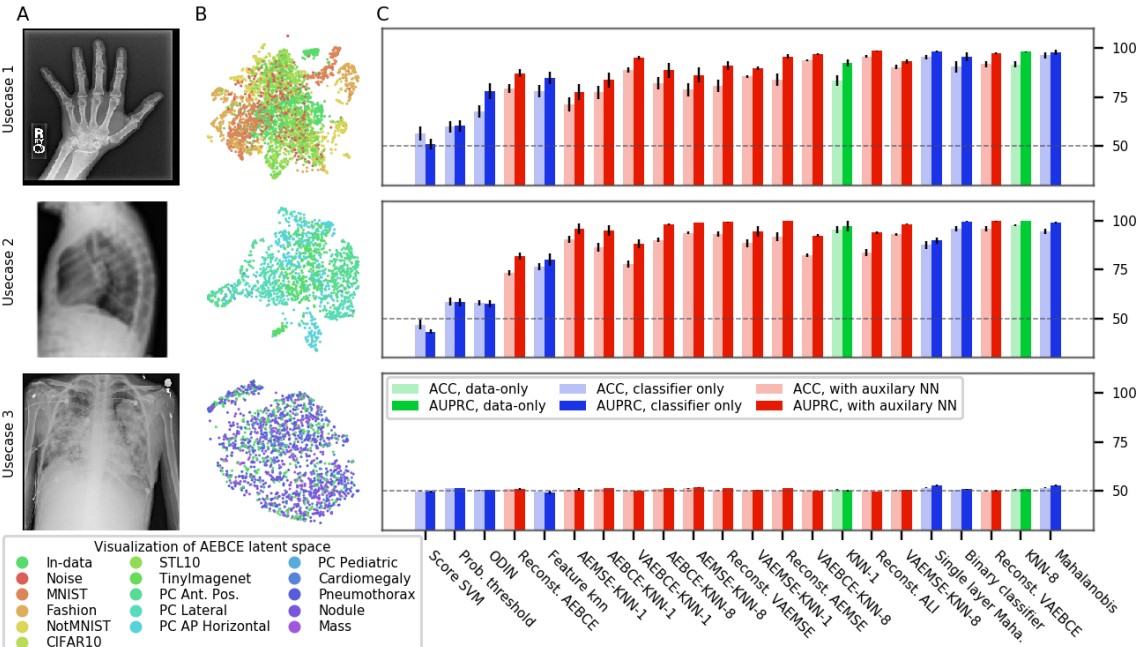

Figure 2: Visualizations and OoDD results on AP view chest-xray (Evaluation 1). Each row of figures correspond to a usecase. Column A shows examples of *Out* data for each usecase (hand x-ray, lateral view chest x-ray, and xray of pneumothorax from top to bottom). Column B shows UMAP visualizations of AE latent space - colors of points represent their respective datasets. Column C plots the accuracy and AUPRC of OoDD methods in each usecase, averaged across all randomized trials. Bars are sorted by accuracy averaged across usecases, and coloured according to method's grouping: green for baseline image space methods, blue for methods based upon the task specific classifier, and red for methods that use an auxilary neural network. Error bars represent 95% confidence interval.

## 5. Experimental Results

Figures 2 through 4, and appendix figure C.1 show the performance of OoDD methods on the four evaluations. Generally, we observe that our choice of datasets for *In* and *Out* data create a range of simple to hard test cases for OoDD methods. While many methods can solve usecase 1 and usecase 2 adequately in evaluations 1-3, usecase 3 proves difficult for all methods tested. This is reflected in the UMAP visualization of the AE latent spaces (column B of figures 2 to 3), in which we observe that the *In* data points are easily separable from *Out* data in usecases 1 and 2, but well-mixed with *Out* data in usecase 3. It is surprising that no method achieved significantly better accuracy than random in usecase 3 of evaluations 1 and 2 across all repeated trials. This illustrates the extreme difficulty of detecting unseen/nouveau diseases, which corroborates the findings of Ren et al. (2019).

**Overall Performance** Across evaluations, the better performing classifier-only methods are competitive with the methods that use auxiliary models. When performance is aggregated across all evaluations (Figure 5), the best classifier-only methods (Mahalanobis and binary classifier) outperform auxiliary models in accuracy. The performance of binary classifier is surprisingly strong. We suspect that this surprising performance is due to the fact that we randomly sample 3 *Out* datasets when constructing $D_{val}$ as opposed to selecting a single *Out* dataset. This added variety in $D_{val}$ *Out* data improves generalization by enforc-

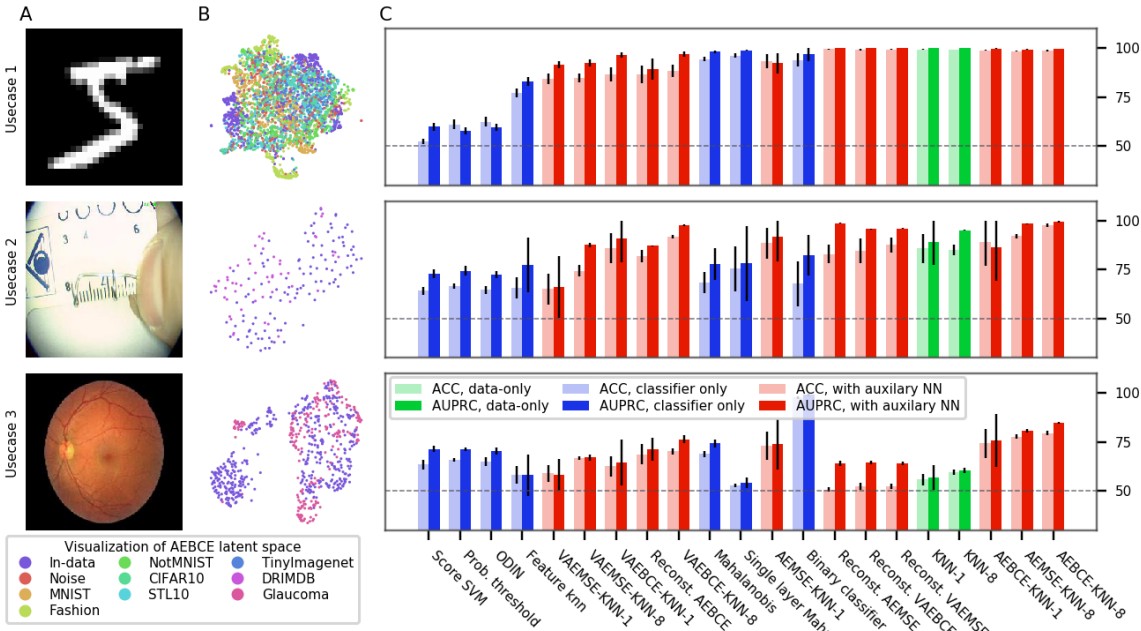

Figure 3: Fundus Imaging (see Figure 2 for description)

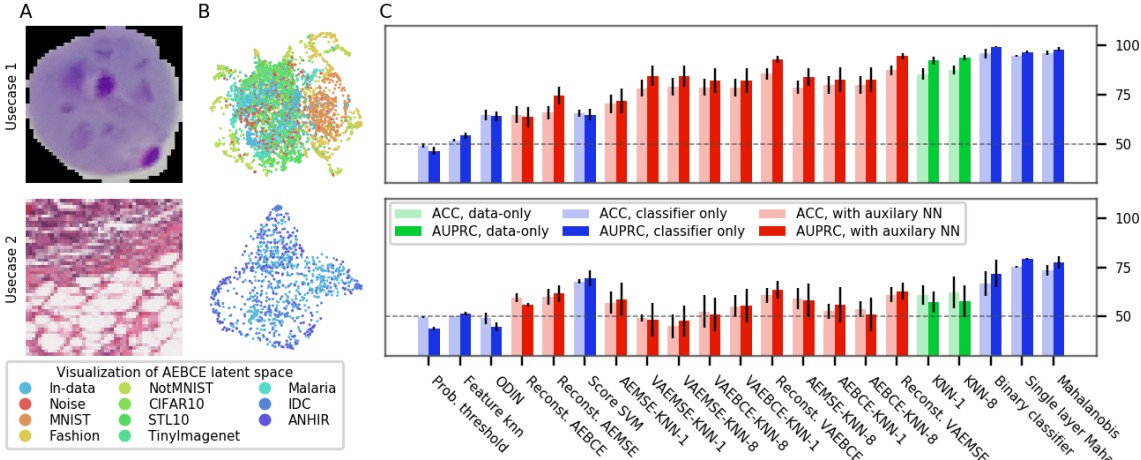

Figure 4: Histology Imaging (see Figure 2 for description)

ing more stable decision boundaries. We performed additional experiments with fewer *Out* datasets on a subset of methods and tasks. Results in figure 7 shows that the gap between the top-4 methods quickly closing with more *Out* datasets in $D_{val}$. The performance of 8 nearest neighbor (KNN-8) is also surprisingly competitive with the best OoDD methods. This may indicate that knowledge of classification on *In* data does not transfer directly to the task of OoDD.

**Accuracy vs. AUPRC as performance metric**   There are some tests with accuracy that's much lower than AUPRC. This is caused by the classification threshold calibrated for $D_{val}$ being ill-suited for classification on $D_{test}$. As AUPRC is computed by scanning all threshold values, it is not effected by the calibration performed on $D_{val}$. If online re-

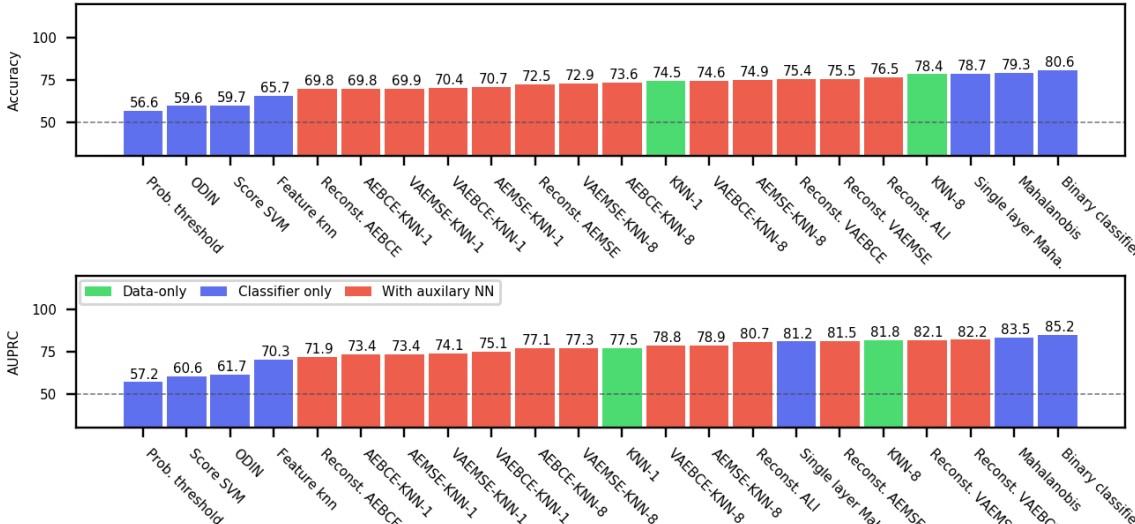

Figure 5: Accuracy and AUPRC of OoDD methods aggregated over all evaluations

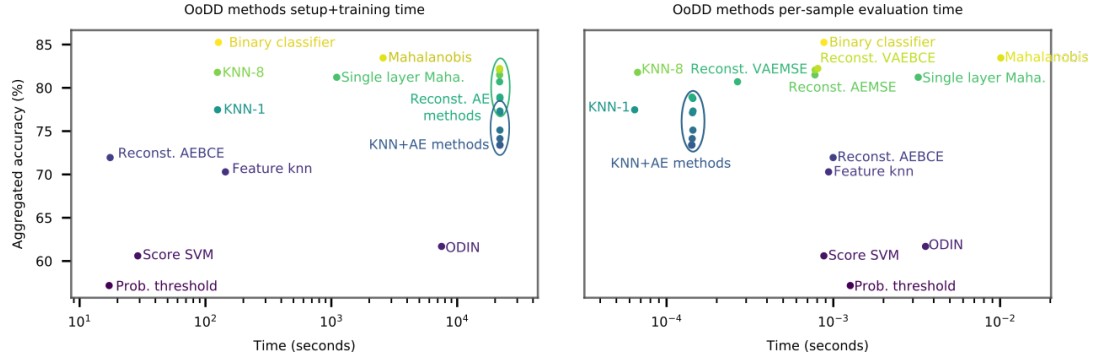

Figure 6: Overall accuracy of methods plotted over total setup time (left) and per-sample run time (right)

calibration is available, then methods with low accuracy and high AUPRC can be improved more significantly over methods with similar accuracy but lower AUPRC.

**Computational Cost** We consider computational cost of each method in terms of setup time and run time. The setup time is measured as the wall-clock computation time taken for hyperparameter search and training. For methods with auxiliary models, the training time of auxiliary neural networks are also included in the setup-time. Run time is measured as the per-sample computation time (averaged over fixed batch size) at test time. Figure 6 plots the accuracy of models over their respective setup and run time. All methods can make predictions reasonably fast, allowing for potential online usage. Mahalanobis and its single layer variant take significantly more time to setup and run than other classifier methods. KNN-8 exhibits the best time vs performance trade-off with its low setup time and good performance. However, as it requires the storage of training images for predictions, it may be unsuitable for use on memory constrained platforms (e.g. mobile) or when training data privacy is of concern.

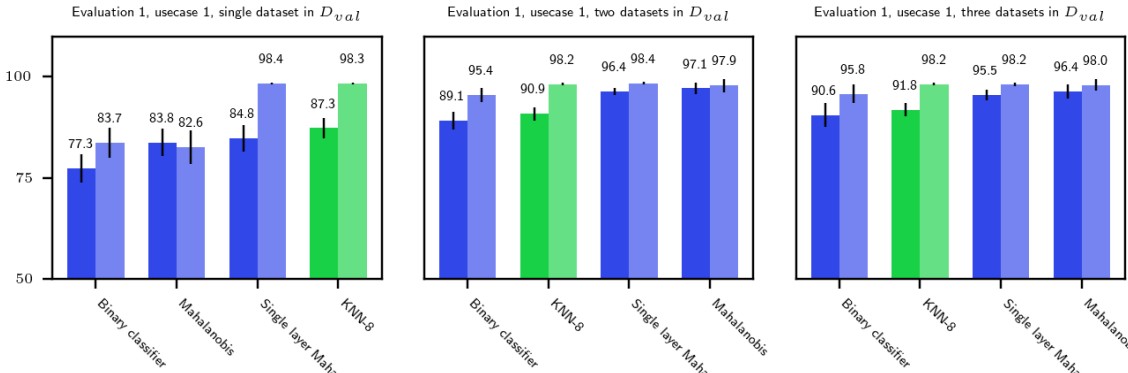

Figure 7: Performance of top-4 methods on frontal X-ray imaging - usecase 1, when trained with fewer datasets in $D_{val}$

## 6. Discussion and Conclusion

Overall, the top three classifier-only methods obtain better accuracy than all methods with auxiliary models except for fundus imaging.

Binary classifier has the best accuracy and AUPRC on average, and is simple to implement. Hence, we recommend binary classifier as the default method for OoDD in the domain of medical images. While usecase 1 and 2 are easily solved with non-complicated models, the failure of most models in almost all tasks to significantly solve usecase 3 is consistent with the finding of Ahmed & Courville (2019). This leaves an open door for future research.

Our findings are almost opposite that of Shafaei et al. (2018), who evaluate on natural images which is a different domain, despite using the same code for overlapping methods. As we performed an extensive hyperparameter on all methods, we conclude that this discrepancy is due to the specific data and tasks we have defined.

In particular, the three worst methods in accuracy and AUPRC are all classifier-only methods. These all perform OoDD on the logits of the pretrained classifier. As our downstream task is binary classification, these methods do not have sufficient information about the images to perform OoDD.

Users of diagnostic tools which employ these OoDD methods should still remain vigilant that images very close to the training distribution yet not in it (and a false negative for usecase 3) could yield unexpected results. In the absence of OoDD methods which have good performance on usecase 3 another approach is to develop methods which will systematically generalize to these examples.

**Limitations** Since we use the downstream task of classifying healthy vs non-healthy for all evaluations, this limits our conclusion to this setting. Other vision tasks such as multiclass classification may provide more useful features and thus see a shift in performance for classifier-based OoDD methods (Zamir et al., 2018). Furthermore, the *In* and *Out* datasets used span many image domains common to medical imaging, but might not be exactly the challenges faced. While we do not intend our selection of datasets to be exhaustive, we justify the choice of the *Out* data by enumerating different types of mistakes or biases that can occur at different stages of the data acquisition, which we refer to as the *uses-cases*.

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

## Appendix A. Dataset details

We use Uniform Noise, Gaussian Noise, CIFAR, MNIST, Fashion MNIST, notMNIST, STL10, and TinyImagenet in the roster of usecase 1 *Out* data for all evaluations. For brevity, this group is denoted as "UC-1 Common".

**Evaluation 1 and 2** To evaluate X-ray task three radiology datasets are used. The first one, **NIH**, is from the NIH Clinical Center in Bethesda, Maryland, USA. This dataset is named the Chest-X-Ray14 dataset (Wang et al., 2017). This dataset contains 108,948 frontal-view X-ray images of 32,717 unique patients with 14 radiological findings. The second, **PC** (Short for PadChest), is from Hospital San Juan de Alicante in Alicante, Spain (Bustos et al., 2019). This dataset consists of 160,000 chest X-rays and reports of over 67,000 patients with frontal and lateral views. The images have been labeled with over 190 different radiological findings, with 27% of the annotations created manually by physicians and the rest extracted from the report by a recurrent neural network. The third is the **MURA** (short for musculoskeletal radiographs) dataset (Rajpurkar et al., 2018) which contains 40,561 bone X-Ray images from 14,863 studies, where each study is manually labeled by radiologists as either normal or abnormal. The X-Ray images contain a finger, wrist, elbow, forearm, hand, humerus, or shoulder.

**Evaluation 3** To evaluate the fundus/retinal (back of the eye) image task we use the **DRD** (Diabetic Retinopathy Detection) [2] dataset from Kaggle which contains 35k fundus images with categorical labels from 0 to 4 indicating the presence of diabetic retinopathy. The **DRIMDB** (Diabetic Retinopathy Images Database) (Sevik et al., 2014) dataset is also used as it contains 216 fundus images labelled as good/bad/outlier. This dataset is specifically designed to assess the quality of fundus images for use in an automated system. Also, the **RIGA dataset** (Retinal fundus images for glaucoma analysis) dataset (Almazroa et al., 2018) provides 460 images which have glaucoma and not diabetic retinopathy.

**Evaluation 4** To evaluate the histology task we start with the **PCAM** dataset (Veeling et al., 2018) (constructed from the CAMELYON dataset (Bejnordi et al., 2017)) which provides patches of H&E stained histology slides of lymph nodes labeled as metastatic (cancerous) or not. The **IDC** (Invasive Ductal Carcinoma) dataset (Janowczyk & Madabhushi, 2016) provides H&E stained images of breast tissue which contains metastatic regions. The **ANHIR** (Automatic Non-rigid Histological Image Registration) dataset (Borovec et al., 2018) provides slide images from multiple tissues including mouse kidney, human lung, and colon stained with a subset of 10 different stains applied to slices in the same region.

## Appendix B. In/Out Data selection details

Evaluation 1's *In* data, the "NIH *In* split", includess all samples in NIH except those labeled with cardiomegaly, nodule, mass, or pneumothorax. Samples with these labels are used in the 'NIH *Out* split' as *Out* data for usecase 3. The PC datasset is subdivided by the view of the chest x-ray. As NIH consists of entirely Posterior-Anterior (PA) views, PC samples of other views - lateral, anterior-posterior (AP), pediatric (PED), and AP horizontal - are

---

2. https://www.kaggle.com/c/diabetic-retinopathy-detection

used as *Out* data for usecase 2. MURA is used as *Out* data for usecase 1 in addition to "UC-1 Common".

Evaluation 2 uses the lateral view images of PC as *In* data. Samples of the same 4 conditions as evaluation 1 are excluded from "PC-lateral *In* split" and placed in "PC-lateral *Out* split". Similar configuration of *Out* data as evaluaiton 1 is used for usecase 1 and 2, except that PC-lateral is substituted by PC-PA.

Evaluation 3 uses DRD as *In* data. The "bad" and "outlier" samples of DRIMDB are used as *Out* data for usecase 2. From RIGA, samples that are positive for glaucoma are used in usecase 3.

Finally, evaluation 4 uses PCAM images as *In* data. Malaria images are used in usecase 1 with "UC-1 Common". To generate *Out* data from ANHIR, square crops of image resolution and magnification equal to PCAM are taken from images full slides. IDC data is used as provided for usecase 2.

In all evaluations , we consider the downstream task of classifying healthy (negative) samples from unhealthy (positive) samples. For NIH, PC-Lateral and PCAM, these labels are already present in the original dataset. In DRD, samples are originally scored for the severity of retinopathy on a scale from 4 (most severe) to 0 (healthy). We consider all samples with scores greater than 0 to be positive.

For each evaluation, $D_{tr}$ is formed by randomly selecting 80% of the *In* data. In usecase 1, $D_{val}$ is sampled from 10% of *In* data and 3 datasets randomly selected from available *Out* datasets of that evaluation. $D_{test}$ is sampled from the remaining 10% *In* data, and *Out* datasets not selected for $D_{val}$. Both $D_{val}$ and $D_{test}$ are sampled for class balance (equal number of *In* and *Out* data). 10 random trials with re-sampled $D_{val}$ and $D_{test}$ are performed for each evaluation. For usecases 2 and 3, $D_{val}$ is formed by the same 10% *In* data. In cases where more than one *Out* dataaest is available (e.g. evaluation 1, usecase 2), one of the *Out* datasets of that evaluation and the remaining are used in $D_{test}$. In this case, we enumerate over all choices of *Out* datasets for $D_{val}$ in each repeated trial. When only one *Out* dataset is available, it is split randomly between $D_{val}$ and $D_{test}$.

## Appendix C. Training details

### C.1. Network training

For classifier models, we use a DenseNet-121 architecture (Huang et al., 2017) with Imagenet pretrained weights. The last layer is re-initialized and the full network is finetuned on $D_{tr}$. As the NIH and PC-Lateral datasets only contain grayscale images, the pretrained weights of features in the first layer are averaged across channels prior to finetuning.

For all of the autoencoders, we use a 12-layer CNN architecture with a bottleneck dimension of 512 for all evaluations. Due to computational constraints, all images are downsampled to $64 \times 64$ when fed to an autoencoder. These AEs are trained from scratch on their respective $D_{tr}$ with MSE loss and BCE loss. We also trained VAEs with the same architectures, except that the bottleneck dimension is doubled to 1024 to allow the code to be split into means and variances.

In addition, we explore the potential benefits of training encoder+decoder using ALI in evaluation 1. We use the same network architecture as proposed in (Dumoulin et al., 2016), with weights pretrained on Imagenet and finetuned on NIH *In* classes. Due to the added

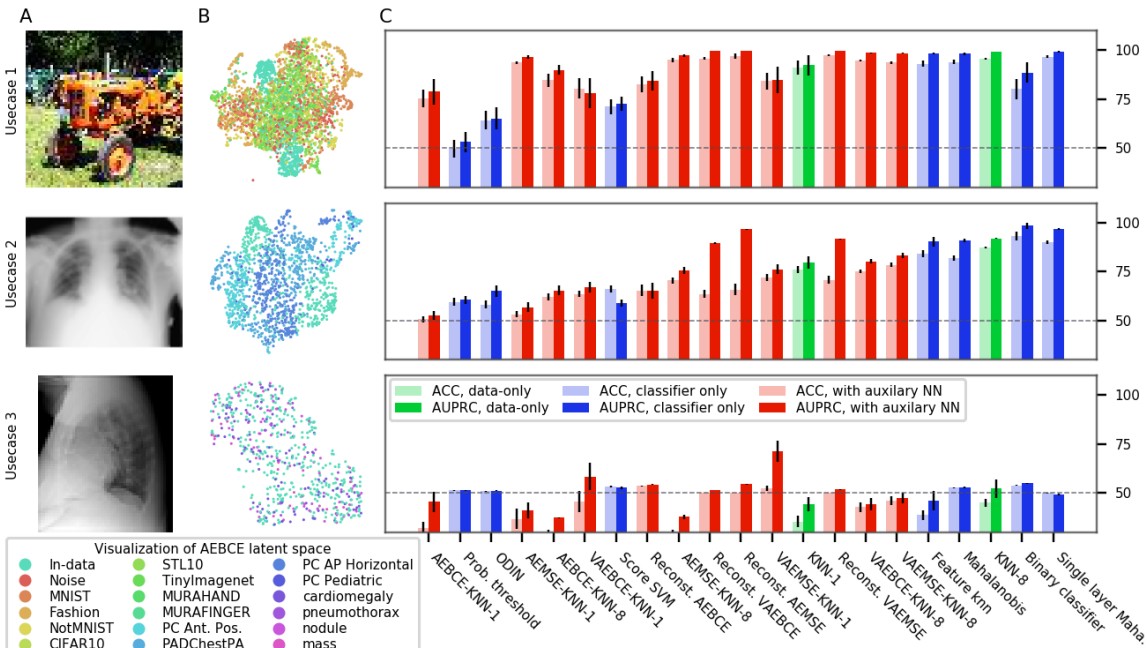

Figure C.1: Lateral X-ray imaging

complexity of training GANs and the lack of significant improvements in OoDD performance over regular AEs (see §5), we did not train ALI models for the other three evaluations.

In order to gauge training progress and overfitting, we hold out 5% of $D_{tr}$ as validation set. We select the training checkpoint with the lowest error on $D_{tr}$ for use in OoDD methods.

## C.2. OoDD methods training

When training for usecase 1, three *Out* datasets are randomly selected for $D_{val}$, while the reset is used for $D_{test}$. These are mixed with the held-out 10% of *In* data (as explained in Appendix B), and then the mixture trimmed such that the number of *In* distribution samples matches that of *Out* samples. Most methods require held-out data to gauge overfitting or calibration. Addtionally, some methods (ODIN and Mahalanobis) require additional hyper-parameter selection. Hence, we further subdivide $D_{val}$ in to a 80% 'training' split and a 20% 'validation' split; methods are trained/optimized on the 'training' split with early-stopping/calibration on the 'validation' split.

