# OpenReview forum: "Which MOoD Methods work? A Benchmark of Medical Out of Distribution Detection"
_MIDL.io/2020/Conference — Submitted to MIDL 2020_

### Official Review · AnonReviewer2 · 2020-03-05
**Good attempt in benchmarking Medical OoD detection**

**Rating:** 3
**Confidence:** 4
**Recommendation:** Poster

**Summary:**

The paper benchmarks OoD detection methods in three separate and commonly used domains of medical imaging. To this end, the paper considers three distinct  OoD task categories and three classes of OoD methods. OoD system is very much essential in the deployment of medical imaging analysis tools in the real world, and this paper tries to establish the benchmark as a systematic evaluation of OoD detection methods for medical imaging application remains absent.

**Strengths:**

1. This paper provides a systematic evaluation of OoD detection methods for medical imaging applications. OoD detection is crucial as the current deep learning systems, owing to their over-confident nature, limits the safe deployment in real-world medical settings.
2. The paper is evaluated on accessible medical imaging datasets (e.g., Chest X-ray), and the experiments are carefully laid out.

**Weaknesses:**

One of the most significant weaknesses of the paper, in my opinion, is the selection of methods for OoD detection. It has been largely known that discriminative networks are over-confident in their prediction, which is why there has been a lot of interest in learning deep generative model-based approaches for OoD detection. Although generative models like VAE have been considered in this work, the lack of analysis of discriminative vs. generative models ignores the recent progress made in the research around OoD benchmarking. It seems the authors cited one such paper [1] but fail to discuss it appropriately in their work.
[1] Do Deep Generative Models Know What They Don't Know? Nalisnick et al., ICLR, 2019.


**Detailed Comments:**

The paper is well written, and I enjoyed reading it. Some minor comments include:
1. The labels in the figure are not very clear and hard to read. There might be some issue with figure quality which the authors can quickly address.
2. Authors have considered using UMAP, a new alternative to widely used t-SNE. Please cite the UMAP paper, which would help acknowledge the contribution of UMAP's authors.
3. The title of the paper might be misleading as the paper is not explicitly pointing out the best models to consider for OoD detection in medical imaging.

**Justification Of Rating:**

The paper is a useful contribution to the medical imaging community. But at the same time, it lacks discussion/experiments on some of the recent developments around OoD benchmarking in the general machine learning domain.

**Paper Type:**

validation/application paper

**Questions To Address In The Rebuttal:**

Is it possible to add an analysis that would compare the generative model to the discriminative models in this work? If this is something too heavy work for the rebuttal period, can the authors at least discuss this in their paper?

**Special Issue:**

no

---

> ### Author Response · Authors · 2020-03-27
> **Response to reviewer 2**
>
> We thank the reviewer for their feedback!
>
> > Is it possible to add an analysis that would compare the generative model to the discriminative models in this work?
> Our results show that the best way to use a VAE for OoD-detection is not through likelihood estimation in the latent space, but rather through comparing the reconstructed image with the input and thresholding the reconstruction loss. This is supported by the analysis in [2] where they related the reconstruction error to an unnormalized estimate of the sample likelihood (marginalized over the latent code). Experimental results in [1] suggest that likelihood estimations given by generative models are not robust scores for OoD detection. Our results have found that classifier based OoDD methods perform better than generative methods which corroborates their observation. We will add this observation to the paper.
>
> > The labels in the figure are not very clear and hard to read. There might be some issue with figure quality which the authors can quickly address.
>
> We will switch to vector graphics for figures in the camera ready version to avert the resolution issue with fonts.
>
> >Authors have considered using UMAP, a new alternative to widely used t-SNE. Please cite the UMAP paper, which would help acknowledge the contribution of UMAP's authors.
>
> We will add this missing citation.
>
> >The title of the paper might be misleading as the paper is not explicitly pointing out the best models to consider for OoD detection in medical imaging
>
> We found that binary classification is the best performing OoDD method overall for medical image data. We will make this recommendation more explicit in the conclusion.
>
>
> [1] Do Deep Generative Models Know What They Don't Know? Nalisnick et al., ICLR, 2019.
> [2] Anomaly scores for generative models. Arxiv preprint, https://arxiv.org/pdf/1905.11890.pdf

---

> > ### Comment · AnonReviewer2 · 2020-04-02
> > **No change in my review or rating.**
> >
> > I thank the authors for their efforts and maintain my initial rating.

---

### Official Review · AnonReviewer1 · 2020-03-11
**Idea is promising but the paper is poorly structured.**

**Rating:** 2
**Confidence:** 5

**Summary:**

The authors explore various methods of detecting out-of-distribution (OOD) samples using several different datasets with varying properties.  They categorise types of OOD samples (type 1: data from a wrong domain, type 2: data from the right domain but e.g. poor quality, type 3: data from the right domain but with previously unencountered abnormality).  They also categorise the various methods from literature of detecting them (data-only, classifier-only, with auxiliary-classifier).  They experiment with chest x-ray, fundus and pathology images and a large number of OOD detection methods.

**Strengths:**

The idea to compare different methods of OOD detection is good and interesting for the community.  The authors made a reasonable effort to include a cross section of data and methods and to analyse the results appropriately.  An interesting finding is that methods using auxiliary classifiers do not perform better.

**Weaknesses:**

As per the subsequent section, the paper is not well structured and is difficult to follow in its current format.  The text requires improvement in clarity, the figures are not well explained and the details of the actual methods implemented are entirely unclear.  Aside from this the results are of mild interest, but since many of the methods perform equally well it is difficult to draw very strong conclusions.

**Justification Of Rating:**

The paper as a whole is difficult to follow and poorly structured.  Important details of the implementations are absent and figures are not easily interpreted or well captioned.  It is difficult to draw very strong conclusions or recommendations from the results.

**Paper Type:**

validation/application paper

**Questions To Address In The Rebuttal:**

The paper is difficult to follow and not well structured or explained.  For example, the downstream classification task is only stated late in the discussion section (and in the appendix), so for much of the paper the reader is unclear what the training task actually is. Another example, there is no explanation anywhere of the 21 methods for which results are provided. Apart from the colour coding, the reader is left to guess what the method was based on the names/acronyms on the X-axis of the bar-charts.  If a reader would want to implement any of these, or understand differences between them that is extremely difficult.   Many terms like 'AEBCE' (for example) are used  in multiple figure legends and never explained in the text.

The authors fail to cite or experiment with the more recent method of Calli et al from MIDL 2019 ("FRODO: Free rejection of out-of-distribution samples: application to chest x-ray analysis")

The selection of the datasets for "Out3" does not make sense to me, for example in the Chest x-ray images, x-rays with nodule or pneumothorax are designated as out-of-distribution since x-rays with these labels are not included in the training set.  However these are typically very small and subtle findings that would not cause the image to have a particularly "out-of-distribution" appearance or affect the ability to classify e.g. pleural effusion or cardiomegaly.  It is not clear to me whether the presence of glaucoma in the fundus images would affect the detection of diabetic retinopathy or not, but this needs to be considered. I would say usecase3 should consist of images with the presence of pathology/artifact that fundamentally alters the ability to diagnose the image correctly - although I appreciate this is difficult to define precisely or to find sufficient data samples. It is not surprising to me that the x-ray experiments failed to identify the out3 images as out-of-distribution.

The UMAP visualizations are not explained and are hard to follow anyway since the colours selected are too close to each other.

The computational cost analysis (figure and large paragraph) could be omitted to make room for more clarity and explanation in the paper.  All methods can be trained (once-off) within a few hours and run at evaluation time of under 0.01 seconds, so more precise times are unlikely to be of interest to practitioners choosing one of these.

Figure 7 is not well explained and the caption is particularly insufficient.  It is unclear to me why changing Dval (which as I understand it is essentially used for setting a good threshold to differentiate In-data from Out-data) would change the area under the curve (the curve which is generated from all possible thresholds).

The over-arching problem in this area is that we do not generally know what the distribution of OUT-data will consist of.  More exploration of how we can build systems robust to all types of OUT-data, including unsee and unexpected types would be useful.

References are incomplete/incorrect.  e.g. Shafaei et al listed as "arxiv 2018" with no arxiv identifier.  Ren et al listed with a title and a year, but no publication information.....etc
Typos, spelling, language errors.


**Special Issue:**

no

---

> ### Author Response · Authors · 2020-03-27
> **Response to reviewer 1**
>
> We thank the reviewer for their feedback!
>
> > Another example, there is no explanation anywhere of the 21 methods for which results are provided.
>
> These methods are discussed in Section 3. "Methods of OOD Detection"
>
> > For example, the downstream classification task is only stated late in the discussion section (and in the appendix)
>
> We discuss the classification tasks in Section 4. "Experiment Setup". As these are well known tasks in the community we didn't think it was necessary to discuss them in detail in the main text but include the details in the appendix.
>
> > The authors fail to cite or experiment with the more recent method of Calli et al from MIDL 2019 ("FRODO: Free rejection of out-of-distribution samples: application to chest x-ray analysis")
>
> We do use this method but we call it Mahalanobis and cite https://arxiv.org/abs/1807.03888 . We will add a reference to the FRODO work as well.
>
> >  It is difficult to draw very strong conclusions or recommendations from the results.
>
> We have shown that a properly tuned simple OoDD method in binary-classifier is the best performing OoDD method. While the gap between binary-classifier and Mahalanobis is small, this conclusion is statistically significant. To the best of our knowledge, our work is the first to quantitatively illustrate the difficulty of detecting images of unseen medical conditions in the context of OoDD. We will state this explicitly in the paper’s conclusion.
>
> > More exploration of how we can build systems robust to all types of OUT-data, including unsee and unexpected types would be useful.
>
> This is the goal of the paper. We don't expect to solve the problem with one paper but we believe this is a step forward and we have done a lot of computational analysis to get these results. Do you have any suggestions on what experiments that we could do to improve this work?

---

> > ### Comment · AnonReviewer1 · 2020-03-30
> > **rebuttal questions not answered in full**
> >
> > The authors have only (partially) addressed a selection of the list of questions which I asked in "questions to address"

---

> > > ### Author Response · Authors · 2020-04-01
> > > **Addressing other points in "questions to address"**
> > >
> > > With respect to the selection of OUT-data in usecase 3, our benchmark is certainly challenging but not unreasonable to expect from OoDD methods. While nodule and mass are difficult to spot in chest x-ray images, cardiomegaly and pneumothorax have visually obvious features. Furthermore, the purpose of OoDD is not solely "condition A's presence affect condition B's diagnosis, hence remove samples with condition A", but rather "the machine learning algorithm has never been trained on samples that have both condition A and condition B, hence it is dangerous to make predictions on B when A is also present."
> > > The computational time (during inference) may be fast in our reported results, but in practice, medical diagnostic tools are either executed on user's personal devices or provided as a cloud service. In the former case, we expect the inference time to increase many folds from our reported times due to difference in computational power. In the later case, the service provider has a strong incentive to select computationally efficient methods to minimize their server load. Hence, we argue that the cost/benefit analysis of OoDD methods are always of practical importance.
> > > With respect to the content of figure 7, Dval is not used for just setting a threshold, rather it is used as Out-data to train the OoD detector. For example, in knn, Dval is directly added to the list of seen samples. We will make this more clear in the discussion section.
> > > We will add a quick explanation of UMAP to the experimental setup section, but since it is a standard visualization method, we will defer interested readers to the original paper [1].
> > >
> > > [1] McInnes, Leland, John Healy, and James Melville. "Umap: Uniform manifold approximation and projection for dimension reduction." arXiv preprint arXiv:1802.03426 (2018).

---

### Official Review · AnonReviewer4 · 2020-03-13
**A poorly written benchmarking paper that does not reveal much new information**

**Rating:** 1
**Confidence:** 2

**Summary:**

The paper aims to report on a benchmark of Out-of-Distribution (OoD) detection methods. Even though this is done with an extensive list of datasets and methods, i find the writing quite poor and hard to follow. It reads like the authors got lost in the mix. At least, that is what happened to me. After all, I do find it dull and cannot recommend for acceptance, especially thinking that it does not tell us something new or interesting.

After all, the following line makes me think that the current work is just an imitation of earlier work:
"Our findings are almost opposite that of Shafaei et al. (2018), who evaluate on natural images  which  is  a  different  domain,  despite  using  the  same  code  for  overlapping  methods."

**Strengths:**

A good ultimate goal to help medical image analysis community. The benchmark involves many methods to compare. A taxonomy of OoD usecases is also given and the authors play with many datasets to design usecases and OoD detection tasks.



**Weaknesses:**

In opinion, the writing of the paper is its biggest weakness. It is hard to follow, which also makes it much harder to understand the results from such a large scale benchmark. Even though the motivation is good, the study fails in clearly reporting its results and merits.

**Detailed Comments:**

"For machine learning (ML) systems,  the expertise is defined by the validation score on the distribution of data used during training, as the performance of the system can be validated on samples drawn from the same distribution (as per PAC learning (Valiant,1984)).   [ This  restriction  can  be  translated  into  the  task  of Out-of-Distribution  Detection(OoDD) ], the goal of which is to distinguish between samples in and out of a desired distribution (abbreviated to In and Out data)."
i do not really understand what the authors mean here by translation. Translation of PAC assumption to OoD problems? If it could be translated so easily, why would we be dealing with OoD methods, problems, etc...?

later --> latter

MOoD, OoDD, OoD, OD: so many variants...

Flow is also a big problem, which persist almost everywhere.



**Justification Of Rating:**

I found it difficult to properly evaluate this paper, due to the aforementioned reasons. I could not recommend a paper for inclusion, since I could not properly understand and judge it. My only suggestion to the authors would be to substantially revise the paper so that they can communicate their work more effectively.

**Paper Type:**

validation/application paper

**Questions To Address In The Rebuttal:**

I found it difficult to properly evaluate this paper, due to the aforementioned reasons. I am afraid to say it but a rebuttal would not change my thoughts.

**Special Issue:**

no

---

> ### Author Response · Authors · 2020-03-27
> **Response to reviewer 4**
>
> We thank the reviewer for their feedback!
>
> > My only suggestion to the authors would be to substantially revise the paper so that they can communicate their work more effectively.
>
> We are very motivated to fix the writing and delivery of the idea. Can you tell us phrases which we should rewrite or a structure that would be better?
>
> > thinking that it does not tell us something new or interesting.
>
> Methods such as K-NN and binary classification has largely been dismissed as too simple to work in recent OoDD literature. We have shown that these simple methods can perform on par with recently proposed methods on medical images given proper tuning. To the best of our knowledge, our work is the first to quantitatively illustrate the difficulty of detecting images of unseen medical conditions in the context of OoDD. We will make these new findings more clear in the paper.
>
> > After all, the following line makes me think that the current work is just an imitation of earlier work
>
> This work addresses a question that currently is not discussed in the literature. How well do OoDD models work on medical data. We use the framework and code from Shafaei et al. (2018), which evaluated on datasets such as MNIST and CIFAR, and extend it to make sense on medical data. Our reader would be a practitioner who would like to deploy an OoDD model (a common attendee of MIDL) as well as a researcher who would like to know about the specific challenges they could work on in OoD-detection applied to medical imagery.
>
> >i do not really understand what the authors mean here by translation."
> OoDD removes data from prediction at test time such that the distribution of testing data matches that of the training data. This allows the iid data assumption of PAC analysis to hold for the downstream classifier, thereby providing performance guarantees at inference time. We can reword this sentence to make it more clear.

---

> > ### Comment · AnonReviewer4 · 2020-04-02
> > **Basically, no change in my review or rating.**
> >
> > As I said in my initial review, the ultimate goal of the study is good but the work fails in writing and presentation. I think we are beating a dead horse here. The work must be substantially revised, which cannot be done in a short rebuttal period. Also, reviewers should not be considered as proofreaders. I think it is the authors' responsibility to ensure a certain level of clarity and quality in a submitted work.
> >
> > I thank the authors for their efforts and maintain my initial rating.

---

> > > ### Author Response · Authors · 2020-04-02
> > > **Clarification on review would be appreciated**
> > >
> > > We are still confused as to which parts do reviewer 4 find unclear in the paper and would love to address these concerns. Logistically, the conference forbids modifications to the paper during the rebuttal period, but allows modifications to be made after the rebuttal until the camera ready deadline, so we are confident in our ability to make necessary changes to prevent reviewer 4 from being lost in the mix.

---

### Official Review · AnonReviewer3 · 2020-03-14
**A useful empirical study comparing methods for detecting medical images that lie 'outside' classifiers' training distributions, on multiple data sets.**

**Rating:** 3
**Confidence:** 4
**Recommendation:** Poster

**Summary:**

An evaluation of several methods for detecting medical images that are 'out of distribution' is presented on multiple medical image datasets. This is an important issue for deployable diagnostic medical image analysis systems. Experiments include three different categories of out-of-distribution data, and three different categories of detection method.

**Strengths:**

A main strength of the paper is the range of methods and datasets used, and the use of different categories (use cases) of out-of-distribution data.
The conclusions clearly point to a need for better methods for detecting images from parts of the domain distribution that are not represented in the training data due to selection bias (e.g. rare diseases).


**Weaknesses:**

The use of AUPRC (area under precision-recall curve) and its interpretation do not always seem justified: the phrase "accuracy that's much lower than AUPRC" seems to imply they are directly comparable; the tasks will only be concerned with a part of the PR curve not all of it, so AUPRC is perhaps not the best choice of metric here.

**Justification Of Rating:**

The authors make a serious attempt to evaluate out-of-distribution methods across a range of medical image scenarios. The results provide useful empirical evidence to inform researchers about the likely relative performance of these methods under different circumstances, as well as pointing to the current limitations of such methods.

**Paper Type:**

validation/application paper

**Questions To Address In The Rebuttal:**

Why use AUPRC, as opposed to ROC analysis?

**Special Issue:**

no

---

> ### Author Response · Authors · 2020-03-27
> **Response to Reviewer 3**
>
> We thank the reviewer for their feedback!
>
> > Why use AUPRC, as opposed to ROC analysis?
>
> We chose to report AUPRC as opposed to AUC-ROC since AUPRC is considered in the context of the population distribution of positives and negatives. In out of distribution detection, we may experience class imbalances where most of the samples are in-distribution. In that scenario, the precision of the predictor is of more importance than the false positive rate. Hence, we think AUPRC is a more natural choice to communicate performance. We will add AUC-ROC of all experiments in the appendix.

---

> > ### Comment · AnonReviewer3 · 2020-04-02
> > **Comment was not fully addressed.**
> >
> > My comment was only partially addressed - the "area under the curve" can be misleading to use as the metric because only a small part of the curve is likely to be relevant for an application.
> >
> > Nevertheless my rating remains unchanged.

---

### Meta-Review · Area_Chair1 · 2020-04-07
**MetaReview of Paper51 by AreaChair1**

**Rating:** 2

**Metareview:**

The initial opinions on the paper were split, with two reviewers suggesting 'Weak accept', one 'Strong reject' and a 'Weak reject'. I did read the paper myself and I share the concerns of reviewers 4 and 1 that it is not very well written and methods are not explained, or very late in the paper. For example the method Mahalanobis is mentioned in the figures, but not in the Methods section, among others. I also find, in addition to some of the reviewers the quality of the rebuttal lacking. For example, reviewer 1 mentions the term AEBCE which is used but never explained, in addition to other methods were the authors just respond that it is in the methods section, where it is not. As such, I lean towards rejection.

**Paper Type:**

validation/application paper

**Special Issue:**

no

---

> ### Author Response · Authors · 2020-04-09
> **Clarification from authors**
>
> In the rebuttal, we mentioned all acronyms in section 3 of the paper, but did not include all combinations of them. For instance, autoencoder (AE) and binary cross entropy (BCE) have been specified, but not autoencoder with binary cross entropy (AEBCE). We will make this more explicit in the figures when referring to this type of combined acronyms by adding a dash (i.e. "AE-BCE").
> Mahalanobis is introduced in the bottom of paragraph on "classifier only methods, page 3. Since the purpose of a benchmark paper is not to re-state every method being tested, we have kept the introduction to each method brief.
> Our work presents a thorough evaluation of many OoDD methods on multiple modes of medical data. Our number of individual experiments (method-data pairing) total over 10000, lending our results with both wide applicability and strong statistical confidence.

---

### Decision · Program_Chairs · 2020-04-11

Reject